# VISION-ON-DEMAND: EFFICIENT VISUAL LANGUAGE UNDERSTANDING WITH INTERMITTENT ATTENTION

## ABSTRACT

Existing approaches for improving the efficiency of Large Vision-Language Models (LVLMs) are largely based on the concept of visual token reduction. This approach, however, creates an information bottleneck that impairs performance, especially on challenging tasks that require fine-grained understanding and reasoning. In this work, we challenge this paradigm by introducing Vision-on-Demand (VoD), a method that reduces inference cost without discarding visual information. Instead of compressing the image, VoD improves efficiency by sparsifying the interaction between image and text tokens. Specifically, the language model attends to the full set of high-resolution visual tokens through a small, strategically placed set of attention layers: general visual context is provided by efficient cross-attention between text-image, while a few well-placed and dynamically selected self-attention layers refine the visual representations themselves, enabling complex, high-resolution reasoning when needed. Based on this principle, we first train a single universal network on a range of computational budgets by varying the number of self-attention layers, and then introduce a lightweight policy mechanism that dynamically allocates visual computation based on per-sample complexity. Extensive experiments show that VoD drastically reduces computational cost while matching or exceeding state-of-the-art results across a diverse suite of benchmarks, and excels in challenging tasks that require detailed visual understanding.

## 1 INTRODUCTION

Large Vision-Language Models (LVLMs) have demonstrated remarkable capabilities in multimodal understanding (Yang et al., 2024; Chen et al., 2024d; Li et al., 2024). These systems typically pair a vision encoder (e.g., CLIP (Radford et al., 2021)) with a large language model (LLM) (Touvron et al., 2023; Jiang et al., 2023; Yang et al., 2024). The vision encoder maps an input image to dense visual tokens, which are passed through a connector module and fed to the LLM alongside the textual prompt/query. Most of LVLM's computations are due to the large number of visual tokens, a cost that increases sharply with image resolution (Li et al., 2024). To mitigate this, a large volume of work has been proposed that explores the idea of *token reduction/compression.* These works reduce the number of visual tokens by dynamically pruning and/or merging redundant tokens at test-time (Arif et al., 2025; Xing et al., 2025; Yang et al., 2025b; Zhang et al., 2024c; Chen et al., 2024b) or by training specialised compressors (Cai et al., 2025; Hu et al., 2024; Chu et al., 2024). While they perform well on tasks requiring coarse visual understanding, we show that they often incur substantial information loss on complex, high-resolution tasks that require fine-grained visual understanding. See accuracy on "easy" vs "hard" in Fig. 1. This is not surprising as such approaches, by shrinking the set of visual tokens, inevitably, create an information bottleneck.

In this work, we propose a completely different and orthogonal path to token compression methods for increasing the efficiency of LVLMs. Unlike prior token reduction/compression methods that aim to reduce the number of visual tokens processed by the LVLM, our approach aims to reduce/sparsify the number of computational layers executed within the LVLM. Specifically, our method strategically executes a limited number of cross-attention and self-attention layers within the LVLM, allowing it to attend and update the full set of visual tokens only at a few selected points during the forward pass. Owing to this property, we coin our method Vision-on-Demand (VoD). Our idea builds upon the observation that the query and answer tokens sparsely interact with the visual tokens (Kaduri et al., 2025) on a select few *critical* layers. A phenomenon that we show to be heavily task-dependent,

with the location and number of layers and the degree of sparsity varying significantly across tasks, depending on those tasks' complexity.

Overall, **we make the following contributions**:

- We start by decomposing the LVLM layer into image-image and text-image (cross-modal) interactions. **Firstly**, we show that executing a fairly small number of cheap cross-attention layers for text-image, that operate on *the same vision representations*, suffice for tasks requiring coarse visual understanding. This alone surpasses prior state-of-the-art methods on a range of vision-language benchmarks in terms of accuracy and speed.

- **Secondly**, we demonstrate that for complex tasks, both prior works and our cross-attention only variant struggle to perform fine-grained visual understanding. We attribute this to the fact that cross-attention layers allow the language tokens to attend to image information but do not update/modify the visual tokens themselves. To alleviate this, we introduce and execute a small number of self-attention layers which perform *update of the visual token*s, enabling a gradual refinement from lower to higher-level visual features.

- **Thirdly**, as different tasks and samples require different amounts of visual detail, we first train a single universal network on a range of computational budgets by varying the number of self-attention layers. Then, we propose an *adaptive inference* by automatically selecting the self-attention layers to be executed on a per-sample basis using a lightweight policy mechanism trained via offline pseudo-labelling.

- **Fourth**, we show that VoD can be *combined with existing token reduction* methods to further improve efficiency without compromising performance.

- **Fifth**, we set a new state-of-the-art on a range of vision language benchmarks, excelling in challenging tasks that require detailed visual understanding. See Fig. 1.

## 2    CLOSELY RELATED WORK

**Efficient LVLMs via Token Reduction:**  To address the computational challenges posed by the large number of visual tokens in LVLMs, several approaches have been proposed to reduce the number of tokens processed by the LLM. These methods can be broadly grouped into two categories: dynamic token pruning and merging techniques (Zhang et al., 2024c;a; Xing et al., 2025; Arif et al., 2025; Shang et al., 2024), and learned token compression strategies (Chu et al., 2024; Yang et al., 2025b; Cai et al., 2025; Hu et al., 2024).

The former category focuses on dynamically identifying the most important tokens, reducing redundancy by pruning or merging the less relevant tokens prior to the LLM (Yang et al., 2025b; Zhang et al., 2024a; Shang et al., 2024), layer-by-layer within the LLM (Xing et al., 2025; Chen et al., 2024b; Yang et al., 2025a; Tan et al., 2025), or both (Zhang et al., 2025a), using heuristic criteria. Examples of criteria include: selecting top-k attended tokens (Chen et al., 2024b), assessing the correlation between patches (Zhang et al., 2024b), using the attention score between image tokens and [CLS] token (Zhang et al., 2024a), rating the vision tokens using the text tokens (Zhang et al., 2024c), or by analysing the information quantity in the attention matrix (Tan et al., 2025). The latter category either replaces the connector module with a learned compressor (Chu et al., 2024), or introduces a new module before the LLM (Yang et al., 2025b) or as part of the vision encoder (Cai et al., 2025; Hu et al., 2024). These methods finetune the LVLM, either fully or partially.

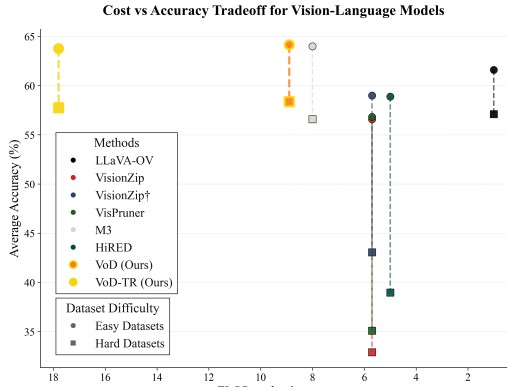

Figure 1: **Efficiency comparison:** FLOPs reduction vs acc. Notice that our approach is significantly more efficient while also retaining the performance on the harder datasets. See Sects. 3 and 5.1 for "easy"-"hard" definition.

While showing promising results, most of these approaches focus on coarser understanding and lower-resolution tasks, often using a LLaVA-1.5 model (Cai et al., 2025; Yang et al., 2025b). Very

few works (e.g., (Arif et al., 2025; Lan et al., 2025)) consider more challenging and fine-grained tasks that require higher resolution, with those that do either suffering from a large accuracy drop (Arif et al., 2025) or exhibiting little to no speed-ups on these datasets (Lan et al., 2025). In this work, we further evaluate existing methods under a unified setting and architecture, and highlight this as a general trend in existing token reduction works. We argue that this performance degradation stems primarily from the information bottleneck inherent in token reduction.

To alleviate this, Vision-on-Demand (VoD) sidesteps the token reduction paradigm altogether. Instead of reducing cost by discarding tokens, VoD strategically limits the number of layers where the language model interacts with and updates visual information, thereby maintaining access to the full, high-resolution visual context throughout the model. This ensures that critical visual details are never permanently lost and can be accessed by the model when needed for fine-grained reasoning, while still achieving significant computational savings. Furthermore, our approach is orthogonal to existing token compression methods and can be combined with them for further efficiency gains.

## 3 MOTIVATION: IMAGE PROCESSING WITHIN LVLMS

To motivate our design, herein, we focus on the internal workings of a standard LVLM (LLaVA-OV) to understand how it utilizes and processes visual information. We analyze the attention patterns of image-image and text-image (cross-modal) interactions and investigate three key questions:

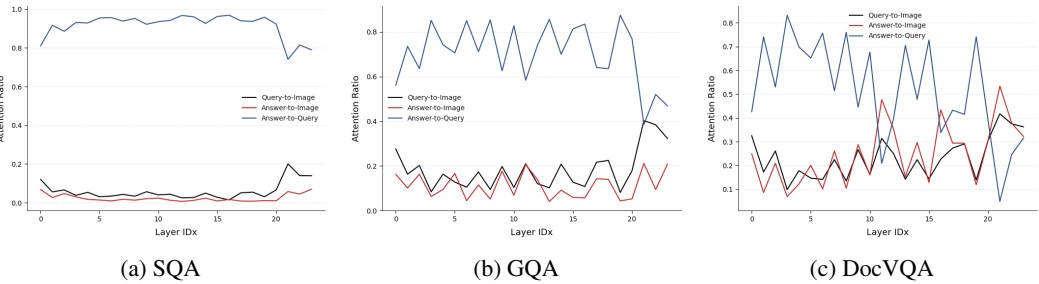

|          (a) SQA          |          (b) GQA          |          (c) DocVQA          |

Figure 2: **Cross-modality attention patterns across layers.** We plot the proportion of attention scores allocated to three interaction types: text queries attending to image tokens (Query-to-Image), answer tokens attending to image tokens (Answer-to-Image), and answer tokens attending to query tokens (Answer-to-Query). For easy tasks like SQA, interaction is sparse and dominated by text-to-text attention. For hard tasks like DocVQA, the model attends to the image across the whole network.

**How often and when does the model look at the image?** We distinguish between three types of interactions: Query-to-Image, Answer-To-Image, and Answer-To-Query. Fig. 2 shows the layer-wise distribution of these interactions for three representative datasets. The results reveal that image-text interactions are task-dependent. For tasks requiring coarse vision understanding (e.g., ScienceQA), the model relies heavily on textual context (Answer-to-Query), with only limited interaction with the image, primarily in the initial and final layers. In contrast, for fine-grained tasks (e.g., DocVQA), the model exhibits sustained attention to the image across the whole network, indicating a continuous need for visual grounding. Moreover, we can observe that critical text-image interactions also occur in the middle layers in addition to the first and last layers. Interestingly, the saw-tooth patterns (for both GQA and DocVQA) suggest that not all cross-attention layers are necessary.

**How do visual representations evolve?** To analyze how vision features evolve across layers within the LLM transformer of the LVLM, we adopt the Centered Kernel Alignment (CKA) (Cortes et al., 2012) similarity metric, following Kornblith et al. (2019); Raghu et al. (2021) (see also Appendix).

We compute the pairwise CKA similarity between vision features from all layers of LLaVA-OV transformer on three representative datasets. As shown in Fig. 3, for easy tasks like ScienceQA, the visual features remain largely unchanged throughout the model (CKA > 0.9), implying that the initial representations are sufficient. However, for hard tasks like DocVQA, the features evolve significantly (CKA drops to 0.6), indicating that the model actively refines visual representations to solve the task. This highlights that while coarse tasks can rely on static visual features, complex tasks benefit from

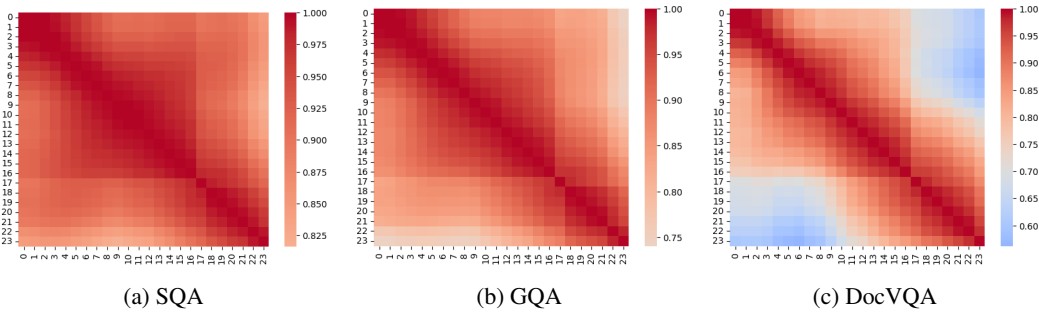

(a) SQA        (b) GQA        (c) DocVQA

Figure 3: **Evolution of visual representations across layers**, measured by pairwise CKA similarity. For easy tasks (e.g., SQA), visual features remain largely static (high similarity across layers). For harder tasks (e.g., GQA and especially DocVQA), features are progressively refined.

the refinement of visual information within the LLM. From the figure, we also observe a series of clusters emerging, indicating that the model refines visual features in stages. The number of stages is task-dependent, and we posit that it indicates the minimum number of self-attention layers that need to be executed to achieve optimal performance.

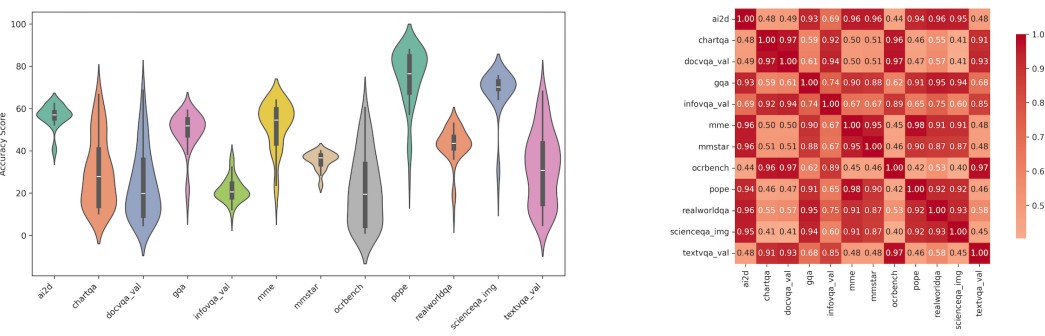

(a) Accuracy distribution per dataset.        (b) Dataset accuracy correlation.

Figure 4: **Accuracy sensitivity by dropping all vision tokens for different subsets of LLM layers.** Left: Accuracy distribution on a dataset-by-dataset basis. Certain datasets (e.g., DocVQA, ChartQA) are particularly sensitive to reduced vision-language interactions. Right: we show how the layer-drop config. & accuracy correlate among datasets. Two clusters emerge: vision-sensitive ("hard") (e.g., InfoVQA, OCRBench, etc) and coarse vision ("easy") (e.g., POPE, SQA, GQA, etc.) datasets.

**What is the impact of reducing image-text interactions?** To this end, we drop *all the vision tokens* from random subsets of LLM layers during inference and measure the performance degradation. Fig. 4 shows that datasets cluster into two groups. "Easy" tasks (e.g., SQA, POPE) are robust to this dropout, maintaining high performance. "Hard" tasks (e.g., DocVQA, ChartQA, InfoVQA) are highly sensitive, with performance dropping sharply as visual processing is reduced. We use this as a basis for dataset categorization in the rest of the paper. This confirms that a one-size-fits-all approach to visual processing is suboptimal; the computational budget should adapt to the sample/task at hand.

**Key takeaways** that inform the design of our proposed Vision-on-Demand (VoD) method: (1) Image-text interactions are sparse, exhibit saw-tooth patterns, and the degree of interaction is highly task-dependent. (2) While coarse tasks can rely on static visual features, complex tasks benefit from dynamic refinement of visual information within the LLM. (3) A one-size-fits-all approach to visual processing is suboptimal; the computational budget should adapt to sample/task demands.

## 4 METHOD

### 4.1 PRELIMINARIES: LARGE VISION-LANGUAGE MODELS

Let $\mathbf{V} \in \mathbb{R}^{N_v \times d}$ and $\mathbf{T} \in \mathbb{R}^{N_t \times d}$ be the sequences of visual and text tokens, respectively, processed by an LVLM. In a standard LVLM, each transformer layer $l$ consists of a self-attention layer followed

by a feed-forward network (FFN) applied to the concatenated sequence $[\mathbf{V}^{(l-1)}; \mathbf{T}^{(l-1)}]$:

$$[\mathbf{V}^{(l)}; \mathbf{T}^{(l)}] = \text{TransformerLayer}_l([\mathbf{V}^{(l-1)}; \mathbf{T}^{(l-1)}]). \tag{1}$$

It is straightforward to observe that the self-attention operating on the concatenated sequence captures all possible image-image, image-text, and text-text interactions. Its computational cost is quadratic in the total sequence length, $O((N_v + N_t)^2 \cdot d)$. Since $N_v \geq N_t$, especially for high-resolution images, the image-image interactions dominate the inference cost.

## 4.2 VISION-ON-DEMAND (VOD)

To reduce the computational cost without performing token reduction, we propose VoD that modifies the LVLM architecture to process visual information sparsely. The core idea is to decouple the processing of text and vision tokens. Most LLM layers operate only on text tokens. Only a few selected layers additionally integrate text-image and image-image interactions by strategically inserting a small number of cross-attention and self-attention layers, as illustrated in Fig. 5 [1]. Crucially, the inserted layers depend on sample/task complexity.

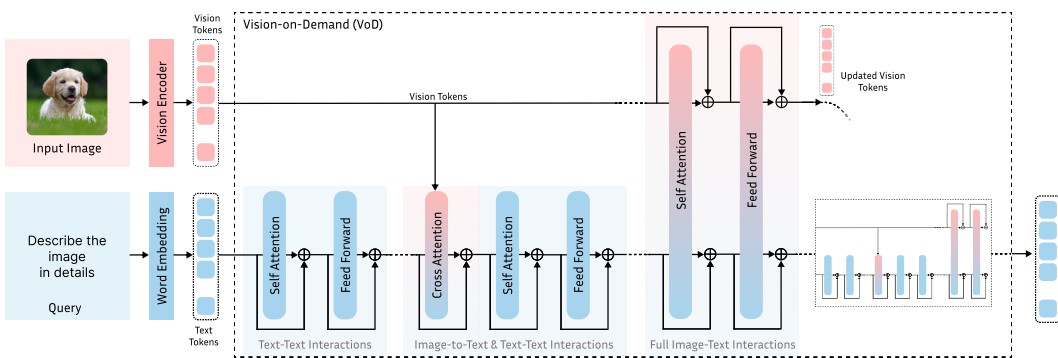

Figure 5: **Conceptual architecture of VoD.** Visual information is sparsely injected into the language stream via a few cross-attention and self-attention layers modelling text-image and image-image interactions. Cross-attention efficiently provides visual context to the text tokens without altering the visual representations. Self-attention, while more costly, refines the visual tokens, enabling subsequent cross-attention layers to access higher-level visual features. This design balances computational efficiency with representational power.

### 4.2.1 EFFICIENT VISUAL CONTEXT VIA CROSS-ATTENTION

For many tasks, the LLM only needs to query visual features without needing to update them. Cross-attention layers provide an efficient mechanism for this, as they integrate visual information into the text processing stream without modifying the visual tokens themselves. We leverage this by having most transformer layers operate solely on text tokens. We then designate a small, uniformly distributed subset of layers, indexed by a set $\mathcal{L}_{CA}$, to perform cross-attention, allowing the text stream to efficiently query the static visual features at selected points.

Let $\mathbf{V}^{(0)}$ be the initial visual tokens from the vision encoder. For a layer $l$, the update rule is:

$$(\mathbf{V}^{(l)}, \mathbf{T}^{(l)}) = \begin{cases} (\mathbf{V}^{(l-1)}, \text{TransformerLayer}_l(\text{CrossAttn}(\mathbf{T}^{(l-1)}, \mathbf{V}^{(l-1)}))), & \text{if } l \in \mathcal{L}_{CA} \\ (\mathbf{V}^{(l-1)}, \text{TransformerLayer}_l(\mathbf{T}^{(l-1)})), & \text{otherwise.} \end{cases} \tag{2}$$

The CrossAttn module uses text tokens as queries and visual tokens as keys and values, and its output is added residually to the text stream. Crucially, in this cross-attention-only variant, visual tokens $\mathbf{V}^{(l-1)}$ are never updated (i.e., $\mathbf{V}^{(l-1)} = \mathbf{V}^{(0)}, \forall l$), making the process highly efficient.

Finally, to ensure the vision tokens retain positional information, which is essential for spatial reasoning, inspired by Chu et al. (2021), we adapt the idea of conditional positional embeddings to 1D sequences and implement them using a 1D depth-wise convolutional layer (with kernel size 7 and a padding of 3). This approach effectively captures both local and global positional information without the slower convergence issues associated with absolute or rotary positional embeddings.

---

[1]More precisely, self-attention models all possible interactions, including the image-image ones.

### 4.2.2 REFINING VISUAL FEATURES WITH SELECTIVE SELF-ATTENTION

The cross-attention only model described in Eq. 2 is efficient and performs well on tasks requiring coarse visual understanding, often surpassing prior state-of-the-art methods. However, the visual tokens remain unchanged, which limits performance on tasks requiring fine-grained reasoning. To address this, we introduce a small number of full self-attention layers on the visual tokens at specific layers, indexed by a set $\mathcal{L}_{SA}$. These layers allow the model to build hierarchical visual representations.

The complete update rule for a layer $l$ becomes:

$$
(\mathbf{V}^{(l)}, \mathbf{T}^{(l)}) = \begin{cases} \text{TransformerLayer}_l([\mathbf{V}^{(l-1)}; \mathbf{T}^{(l-1)}]), & \text{if } l \in \mathcal{L}_{SA} \\ (\mathbf{V}^{(l-1)}, \text{TransformerLayer}_l(\text{CrossAttn}(\mathbf{T}^{(l-1)}, \mathbf{V}^{(l-1)}))), & \text{if } l \in \mathcal{L}_{CA} \\ (\mathbf{V}^{(l-1)}, \text{TransformerLayer}_l(\mathbf{T}^{(l-1)})), & \text{otherwise.} \end{cases}
\tag{3}
$$

When $l \in \mathcal{L}_{SA}$, a standard transformer layer processes both visual and text tokens, updating $\mathbf{V}^{(l-1)}$ to $\mathbf{V}^{(l)}$. Subsequent cross-attention layers ($l' > l, l' \in \mathcal{L}_{CA}$) will then use these refined visual tokens $\mathbf{V}^{(l)}$, enabling more effective context integration. In practice, we find that distributing a small number of cross-attention and self-attention layers uniformly across the model yields strong performance.

### 4.2.3 TRAINING A UNIVERSAL MODEL WITH ON-DEMAND COMPUTE BUDGET

A key insight from our analysis in Sec. 3 is that different tasks require varying amounts of visual processing. To accommodate this, VoD is designed to support dynamic adjustment of the attention layers executed. To this end, we propose the following training strategy:

1. First, we identify the configuration that achieves the best overall performance across all tasks. This configuration determines the maximum number of cross-attention and self-attention layers to be used. Empirically, we find that setting $|L_{CA}| = \frac{1}{3}L$ and $|L_{SA}| = \frac{1}{3}L$ works well in practice, preserving the full model's performance. We initially train the model with all these layers enabled.

2. Secondly, we choose the $L_{CA}$ cross-attention layers to be always executed as they are cheap [2] and provide essential visual context. We choose to vary only the number of self-attention layers. Specifically, we evaluate the model's performance by systematically varying the number of self-attention layers from 0 to $|L_{SA}|$, testing various subsets of the $|L_{SA}|$ layers. This step helps to identify viable configurations that maintain high accuracy at different budgets.

3. Inspired by Cai et al. (2019), we finetune the model by randomly selecting one of these viable configurations during training. This results in a universal model that works robustly for any of the configurations used during training, and hence, across a wide range of computational budgets.

### 4.3 ADAPTIVE INFERENCE

As highlighted in Sec. 3, the amount of visual processing required varies significantly depending on the task and even across individual samples within the same benchmark. This observation indicates that a single, fixed configuration may not be optimal for all scenarios. To address this, we utilize our universal model of Sec. 4.2.3 (designed to operate across a range of pre-defined computational budgets) and introduce a lightweight policy network that dynamically decides how many self-attention layers to execute for each input, enabling per-sample adaptation.

We implement this with an internal routing mechanism. A special *routing* token is appended after the question, and we place an MLP layer at the block prior to the first self-attention block that is a candidate for being skipped. That MLP processes the routing token and predicts the optimal configuration for the subsequent self-attention layers. If multiple questions are present, the model conservatively selects the configuration with the highest computational cost among the individual predictions to ensure sufficient processing capacity.

Since training a routing mechanism can be unstable, we adopt an offline pseudo-labeling approach. First, we run our universal model on a training subset, logging the correctness and token-level losses for each potential layer configuration. We then generate a pseudo-label for the subset by identifying the most efficient configuration. To do this, we first filter for configurations that achieve at least 99%

---

[2] The FLOPs for a full self-attention layer (attention + MLP) are approx. $O((N_t + N_v)^2 d + (N_t + N_v)d^2)$, whereas, for a cross-attention layer, they are only $O(N_t N_v d)$.

of the full model's accuracy. From this group, we select the one with the fewest layers and the lowest aggregate loss. This chosen configuration becomes the target label for training the policy network using a standard cross-entropy loss.

### 4.4 COMBINING VISION-ON-DEMAND WITH TOKEN REDUCTION

Our approach is orthogonal to existing token reduction methods and can be combined with them for further efficiency gains. To this end, we explore two strategies: (i) combining VoD with top-performing token pruning methods (Yang et al., 2025b; Zhang et al., 2025b), and (ii) designing a simple token packing strategy that works with arbitrary token compression ratios. The latter method is coined VoD-TR. Additional details can be found in the Appendix.

## 5 EXPERIMENTS

We compare our method against state-of-the-art approaches on a wide range of vision-language benchmarks, covering tasks that require both coarse and fine-grained visual understanding. We show that prior methods are competitive on easy tasks, but struggle on harder tasks that require detailed visual reasoning. In contrast, our method consistently outperforms prior works across all benchmarks, particularly excelling on the challenging tasks.

### 5.1 EXPERIMENTAL SETUP

**Model architecture and training details:** We build upon the open-sourced LLaVA-OV model (Li et al., 2024), which uses a SigLIP-400M (Zhai et al., 2023) vision encoder, a Qwen2 (Yang et al., 2024) LLM, and a 2-layer MLP connector. The vision encoder operates on $384 \times 384$ image patches, each patch resulting in 729 visual tokens. We insert cross-attention and self-attention layers uniformly across the LLM, at a maximum of 1/3 of the total layers each. We train our model on the same datasets as LLaVA-OV, i.e., (1) the 4M pretraining knowledge data formed by combining synthetically labeled parts of CC3M (Sharma et al., 2018), COCO118K (Lin et al., 2014), BLIP558K (Liu et al., 2024a), SynthDog (Kim et al., 2022) and Evol-Instruct (Chen et al., 2024a) and (2) the LLaVA-OV Single-Image 3.2M dataset (Li et al., 2024), a high-quality mixture formed by combining over 80 datasets. We note that some of the partitions were not made available, hence, in practice, we train on a smaller set (as defined in the LLaVA-OV GitHub repository).

Our training follows a similar two-stage procedure. First, we finetune the new attention layers on the 4M knowledge dataset while freezing the rest of the model. Then, we finetune the entire model on the 3.2M high-quality dataset. Training spans 3 epochs across these stages, using the AdamW (Loshchilov, 2017) optimizer with no weight decay, a batch size of 128, and learning rates of $1e - 4$ and $1e - 5$ for the first and second stages, respectively. We train on 16 MI300X GPUs using PyTorch (Paszke et al., 2019) and DeepSpeed (Rasley et al., 2020). This process applies to both universal and independent variants, except that the universal model also samples configurations during training (Sec. 4.2.3, and appendix for details).

**Vision-language benchmarks:** We evaluate our models on a comprehensive set of vision-language benchmarks designed to assess diverse aspects of visual understanding. Specifically, we include the following datasets: RealWorldQA (xAI, 2024), ScienceQA (Lu et al., 2022), GQA (Hudson & Manning, 2019), MME (Zhang et al., 2021), MMSTAR (Chen et al., 2024c), MMBench (Liu et al., 2024c), POPE (Li et al., 2023), AI2D (Kembhavi et al., 2016), ChartQA (Masry et al., 2022), TextVQA (Singh et al., 2019), InfoVQA (Mathew et al., 2022), OCRBench (Liu et al., 2024d), and DocVQA (Mathew et al., 2021). To better analyze the model's performance, based on the observations from Sec. 3, we categorize these datasets into two: *easy*, which involve limited text-to-image interactions, and *hard* tasks, requiring extensive text-to-image and image-to-image interactions.

**State-of-the-art baselines:** We compare our method against several training-free and training-aware approaches. Since many existing token reduction methods are designed for different architectures (e.g., LLaVA-1.5 (Liu et al., 2024a)) and focus on lower-resolution tasks, we re-implement and

evaluate them under a unified setting using the same LLaVA-OV architecture and, where applicable, training data. Details of the re-implementations are provided in the Appendix.

## 5.2 COMPARISON WITH THE STATE-OF-THE-ART

We compare our method against state-of-the-art approaches using a shared LLaVA-OV (0.5B) backbone. In addition to the numerical accuracy, we also report average FLOP savings relative to the baseline LLaVA-OV model. Note that we do not take into consideration the vision encoder FLOPs as they are common to all methods. For our method, we always use a single universal VoD model that ensures adaptability across different tasks. Table 1 summarizes the results.

VoD achieves significant improvements in both accuracy and computational efficiency. On tasks requiring coarse visual context, VoD matches or exceeds the performance of prior methods while achieving up to $8.6\times$ FLOP savings. For tasks demanding fine-grained visual reasoning, VoD outperforms all baselines, including token reduction methods like VisionZip (Yang et al., 2025b), HiRED (Arif et al., 2025), and $M^3$(Cai et al., 2025), which struggle with information bottlenecks. When combined with token reduction techniques (VoD-TR) in Sec. 4.4, our method achieves even greater efficiency (up to $18\times$ FLOP savings), while maintaining state-of-the-art accuracy.

See the appendix for more results, including comparisons using larger backbones (LLaVA-OV 1.5B).

Table 1: Comparison with state-of-the-art methods on various vision-language benchmarks. The metric used is accuracy for all datasets, except for MME where we report a score (higher is better; MME values are divided by 20 for normalization purposes).

| Method | Easy | | | | | | | | | Hard | | | | | Avg. FLOPs Savings |
| | RWorldQA | SQA | GQA | MME | MSTAR | POPE | TextVQA | AI2D | Avg. (Easy) | ChartQA | OCRBench | InfoVQA | DocVQA | Avg. (Hard) | |
| --- | --- | --- | --- | --- | --- | --- | --- | --- | --- | --- | --- | --- | --- | --- | --- |
| LLaVA-OV (Li et al., 2024) | 54.0 | 67.2 | 58.3 | 60.6 | 40.6 | 88.4 | 66.0 | 56.7 | 61.5 | 60.9 | 58.8 | 40.0 | 68.7 | 57.1 | 1.0× |
| VisionZip (Yang et al., 2025b) | 51.9 | 67.0 | 53.0 | 62.1 | 38.9 | 85.5 | 46.5 | 53.2 | 57.3 | 44.0 | 27.0 | 23.9 | 36.7 | 32.9 | 5.7× |
| VisionZip† (Yang et al., 2025b) | 54.7 | 65.8 | 55.7 | 61.9 | 39.3 | 86.9 | 55.7 | 54.8 | 59.3 | 51.2 | 45.0 | 27.2 | 48.8 | 43.1 | 5.7× |
| VisPruner (Zhang et al., 2025b) | 53.3 | 65.7 | 53.2 | 60.1 | 38.1 | 85.9 | 47.8 | 53.7 | 57.2 | 44.0 | 28.3 | 25.4 | 42.7 | 35.1 | 5.7× |
| PyramidDrop (Xing et al., 2025) | 53.1 | 66.7 | 53.5 | 59.4 | 40.1 | 86.0 | 45.5 | 54.3 | 57.3 | 51.1 | 42.2 | 30.4 | 48.0 | 42.9 | 4.2× |
| $M^3$ (Cai et al., 2025) | 54.0 | 75.1 | 59.7 | 63.8 | 40.9 | 88.6 | 67.0 | 62.5 | 64.0 | 64.7 | 58.0 | 38.3 | 65.4 | 56.6 | 8.0× |
| HiRED (Arif et al., 2025) | 52.6 | 66.3 | 55.4 | 61.9 | 39.2 | 86.6 | 56.8 | 55.3 | 59.3 | 47.5 | 33.9 | 26.1 | 48.4 | 39.0 | 5.0× |
| VoD (**Ours**) | 54.6 | 75.3 | 61.8 | 60.5 | 40.1 | 87.6 | 67.8 | 61.5 | 63.6 | 65.2 | 61.8 | 37.6 | 68.9 | 58.4 | 8.6× |
| VoD-TR (**Ours**) | 55.4 | 75.4 | 60.7 | 59.5 | 38.5 | 87.7 | 67.4 | 61.9 | 63.3 | 65.3 | 60.7 | 37.4 | 67.7 | 57.8 | 18× |

# 6 ABLATION STUDIES AND ANALYSIS

Unless otherwise specified, all ablation studies are performed using the LLaVA-OV (0.5B) backbone, trained on the same datasets as the main experiments. The results reported are aggregated across the two task categories for brevity.

Table 2: Accuracy comparison when combining VoD with token reduction methods.

| Method | FLOPs sav. | Easy | Hard |
| --- | --- | --- | --- |
| VoD | 8.9× | 63.6 | 58.4 |
| VoD-TR [2x] | 17.8× | 63.3 | 57.8 |
| approx. 4x token reduction | | | |
| VoD-TR [4x] | 35.0× | 63.1 | 56.2 |
| VoD + VisionZip | 37.0× | 63.3 | 55.3 |
| VoD + VisPruner | 39.0× | 63.5 | 55.9 |

Table 3: Accuracy comparison across configurations and categories.

| SA | CA | Easy | Hard |
| --- | --- | --- | --- |
| 0 | 6 | 63.3 | 51.8 |
| 0 | 8 | 63.4 | 52.0 |
| 0 | 10 | 63.5 | 52.1 |
| 2 | 8 | 63.2 | 55.1 |
| 4 | 8 | 63.3 | 56.9 |
| 7 | 8 | 63.9 | 58.3 |

**Effect of cross-attention and self-attention layers:** Herein, we analyze the impact of varying the number of cross-attention (CA) and self-attention (SA) layers on accuracy. To avoid a potential sampling bias, each configuration corresponds to an independently trained model. From Table 3, we can observe that: (1) cross-attention alone suffices for tasks requiring coarse visual context, with performance saturating around 8 layers; (2) Cross-attention alone is insufficient for tasks demanding

fine-grained reasoning, significantly lagging behind the full model; (3) adding self-attention layers substantially boosts performance on fine-grained tasks, with a 7 layer configuration nearly matching the full model. This underscores the need for visual feature refinement in complex tasks.

**Combining VoD with token reduction methods:** Our method is orthogonal to token reduction techniques and can complement them for greater efficiency. We evaluate the impact of combining VoD with token reduction methods like VisionZip (Yang et al., 2025b), VisPruner (Zhang et al., 2025b), and our token packing strategy (VoD-TR) from Sec. 4.4, under varying reduction rates. As Table 2 shows, our approach can benefit from token reduction, achieving up to $35\times$ FLOPs savings with only a minor drop in accuracy. Notably, more aggressive token reduction rates (e.g., $4\times >$) lead to larger performance drops on hard tasks, as the information bottleneck becomes more pronounced.

**Independent vs universal model training:** Our final model is trained to support multiple configurations, enabling dynamic adjustment of computational cost during inference. Herein, we analyze the performance trade-offs of this universal training approach compared to independently training models at a few selected budgets. As Table 4 indicates, the universal model matches and, surprisingly, surpasses the performance of independently trained models across different budgets, while providing the flexibility of adaptive inference. This suggests that the universal training approach also acts as a form of regularization, improving generalization across configurations.

**Efficiency analysis:** We analyze the computational efficiency of VoD by measuring the number of floating-point operations (FLOPs) required for inference. The primary source of savings in our method comes from replacing the expensive full self-attention over all tokens with either text-only self-attention or a cheaper cross-attention mechanism in most layers. The cost of a standard transformer layer is quadratic in the total sequence length, $O((N_t + N_v)^2 d + (N_t + N_v)d^2)$, which is dominated by the large number of visual tokens $N_v$. In contrast, our cross-attention layers have a cost of only $O(N_t N_v d)$, and text-only layers are independent of

Table 4: Accuracy comparison between independently trained models and a universally trained model supporting multiple configurations. Both model variants use the same fixed configuration for all samples.

| SA | CA | Universal | Easy | Hard |
|----|----|-----------|------|------|
| 2  | 8  | ×         | 63.1 | 55.1 |
|    |    | ✓         | 63.8 | 56.3 |
| 7  | 8  | ×         | 63.6 | 58.3 |
|    |    | ✓         | 64.2 | 59.1 |

$N_v$. The full, expensive self-attention is computed only in a small, selective subset of layers.

Fig. 1 illustrates the computational cost - accuracy tradeoff. Our approach significantly reduces the FLOPs compared to the baseline LLaVA-OV model while offering a better accuracy-efficiency trade-off compared to all prior methods. Note that the FLOPs measured here only account for the transformer layers, excluding the vision encoder, which is common to all methods.

We also measure actual inference speedups on real hardware (MI300X GPUs). A full LLaVA-OV model takes 0.0738sec/sample, a token pruning solution (i.e., VisionZip, VisPruner) at 8x reduction factor - 0.0274sec/sample, while VoD takes 0.0384sec/sample at a 8CA-7SA configuration and 0.0261sec/sample at a 8CA-2SA. The numbers are reported at the max batch size that allows all methods to fit in memory. We observe a reasonable correlation between actual speedup and FLOP savings.

# 7 CONCLUSION

In this work, we proposed Vision-on-Demand (VoD), a method that reduces the inference cost in LVLMs by sparsifying the image-text and image-image interactions without discarding visual information (as in previous token reduction methods). VoDuses efficient cross-attention to model text-image interactions and few selective self-attention layers for visual feature refinement necessary for fine-grained visual understanding and reasoning. VoDtrains a single universal network on a range of computational budgets and then, during inference, utilizes lightweight policies that dynamically allocate visual computation based on per-task or per-sample complexity. We show that VoD drastically improves efficiency, and outperforms state-of-the-art token compression methods across a wide range of benchmarks, especially for challenging tasks that require detailed visual understanding.

## REPRODUCIBILITY STATEMENT

We have made significant efforts to ensure the reproducibility of our work. The implementation details of our proposed Vision-on-Demand (VoD) method, including the architecture modifications, training procedures, and evaluation protocols, are described in detail in Sections 4 and 5 of the main paper. Additionally, we provide a comprehensive set of ablation studies in Section 6 and the appendix to validate the design choices. The datasets used in our experiments are publicly available.

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

# A APPENDIX

## A.1 IDENTIFYING PROMISING CONFIGURATIONS FOR ADAPTIVE TRAINING AND INFERENCE

Considering a model with $L_{SA}$ self-attention layers, we can create $2^{L_{SA}}$ different configurations by choosing to execute or skip each self-attention layer. This results in a vast configuration space, making it somewhat impractical to evaluate all of them. Moreover, many configurations may lead to catastrophic performance degradation, as they may skip critical layers needed for certain tasks. Hence, to facilitate the training process, we seek to identify a subset of promising configurations that maintain high performance. This subset can then be used for adaptive training and inference.

Figure 6 visualizes the performance of a representative subset of configurations on different datasets, with each row representing a configuration and each column a dataset. The color intensity indicates the relative accuracy achieved by that configuration on the respective dataset. From this visualization, we can identify that: (1) dropping the 1st layer leads to significant performance degradation across all datasets, indicating its critical role; (2) configurations with very few self-attention layers (e.g., 1 or 2) perform poorly on complex tasks, while those with more layers generally yield better results; (3) less vision intensive tasks generally prefer a configuration close to early-exit while more complex tasks benefit from a uniform distribution of self-attention layers.

Based on these observations, for a 0.5B model, we subsequently select the following configurations, where each number denotes the location at which a self-attention layer is executed for the vision tokens: [1,4], [1,7], [1,4,7], [1,4,16], [1,7,16], [1,10,16], [1,4,7,16], [1,4,7,22], [1,4,10,16], [1,4,7,10,16], [1,4,7,16,22], [1,4,7,10,16,22], [1,4,7,10,16,19,22].

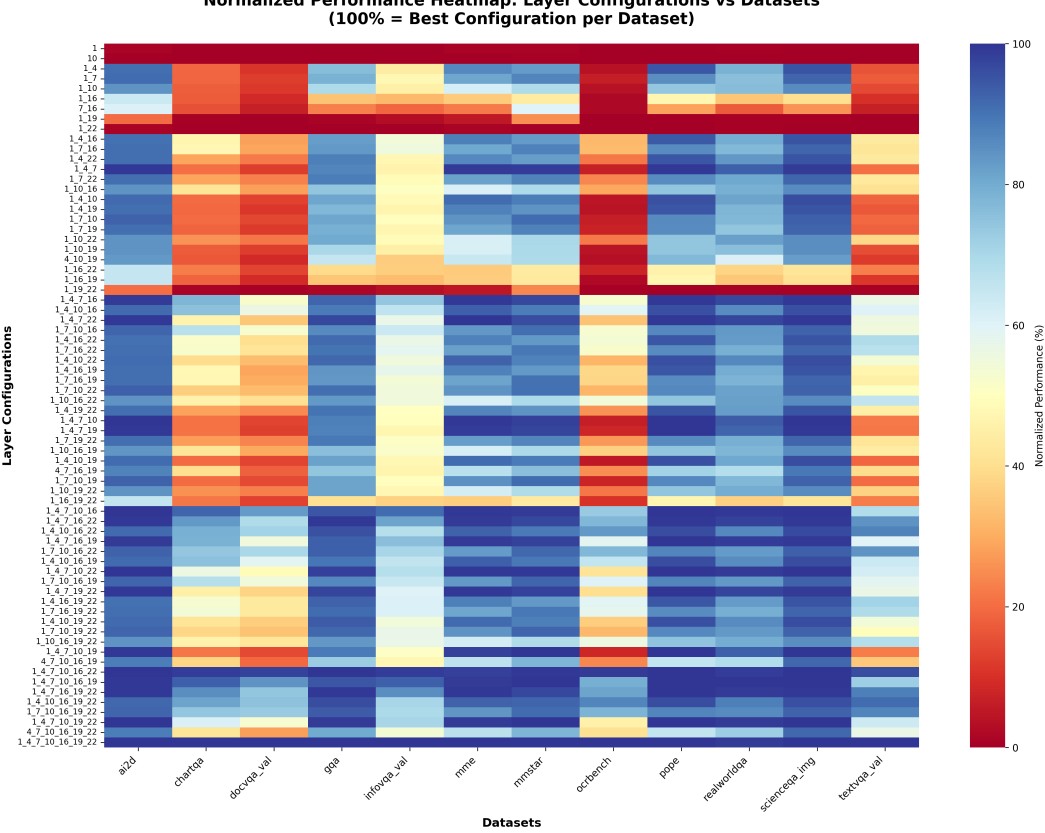

Figure 6: Performance heatmap for different configurations across datasets. Each row represents a configuration, and each column corresponds to a dataset. The color intensity indicates the relative accuracy achieved by that configuration on the respective dataset.

## A.2 PER-DATASET SAVING RATE

In the main manuscript, for brevity, we report the computational savings aggregated across all datasets. Herein, we provide a more detailed per-dataset analysis. Table 5 summarizes the results. For each variant, the top row indicates the accuracy, while the bottom row shows the FLOPs savings relative to the baseline LLaVA-OV model. The same general pattern holds: easy tasks can be solved with very few self-attention layers, while hard tasks require more layers for optimal performance. Our router is able to correctly identify this trend.

Table 5: Per-dataset saving rates. We compare our method against state-of-the-art approaches using a shared LLaVA-OV (0.5B) backbone. For each method, the top row indicates the accuracy, while the bottom row shows the FLOPs savings relative to the baseline LLaVA-OV model. The metrics used are accuracy for most datasets, except for MME where we report a score (higher is better).

| Method | RWQA | SQA | GQA | MME | MMSTAR | POPE | TxtVQA | AI2D | CQA | OCRB | InfoVQA | DocVQA |
|--------|------|-----|-----|-----|--------|------|--------|------|-----|------|---------|--------|
| VoD | 54.6 | 75.3 | 61.8 | 60.5 | 40.1 | 87.6 | 67.8 | 61.5 | 65.2 | 61.8 | 37.6 | 68.9 |
| | 8.4× | 8× | 12× | 8.3× | 10× | 12× | 6.3× | 12× | 8× | 6.6× | 6× | 6× |
| VoD-TR | 55.4 | 75.4 | 60.7 | 59.5 | 38.5 | 87.7 | 67.4 | 61.9 | 65.3 | 60.7 | 37.4 | 67.7 |
| | 24× | 16× | 24× | 17.7× | 17.7× | 24× | 14.5× | 24× | 16× | 15× | 12× | 12× |

To provide further insight into the routing mechanism's behaviour, we present in Figure 7 the layer configurations it chooses for each test set dataset. Two significant observations can be made from this figure: a) the routing mechanism is largely consistent with regard to the computational budget allocated for each dataset, as the configurations chosen for each dataset tend to have a similar number of layers, and b) despite the fact that the original labels are defined in a per-dataset basis, the routing mechanism indeed operates on a per-sample basis, which makes it adaptive to individual samples' complexity.

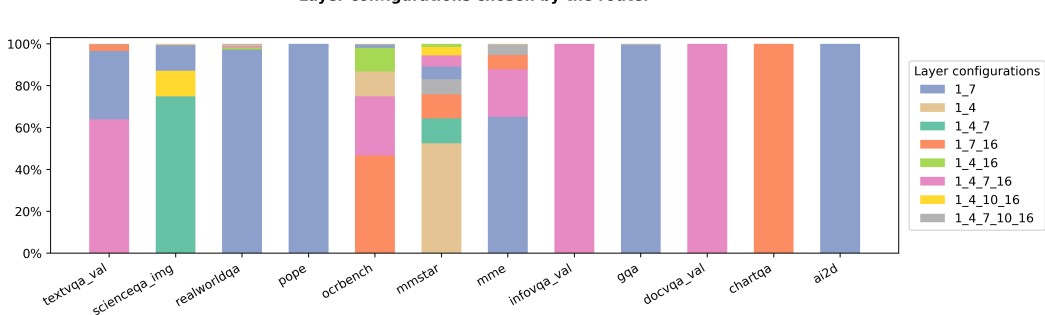

Figure 7: Layer configuration assignments made by the router for each test dataset.

## A.3 PERFORMANCE ACROSS ALL INDIVIDUAL CONFIGURATIONS

Our universal model is trained by randomly sampling a viable configuration during training. To evaluate the effectiveness of each configuration, Figure 8 illustrates their performance across all downstream tasks. On the easy partition, most configurations achieve similar performance regardless of their computational cost. In contrast, for challenging tasks, performance improves almost linearly with the computational budget, highlighting once more the importance of additional self-attention layers for fine-grained reasoning.

## A.4 ORACLE PERFORMANCE ANALYSIS

To assess the potential of our adaptive inference mechanism, we conduct an oracle analysis where we select the optimal configuration for each sample from our predefined set. Figure 9 illustrates the distribution of the selected configurations across all samples such that the overall accuracy is maximized. These results reinforce the conclusion that most samples can be accurately processed

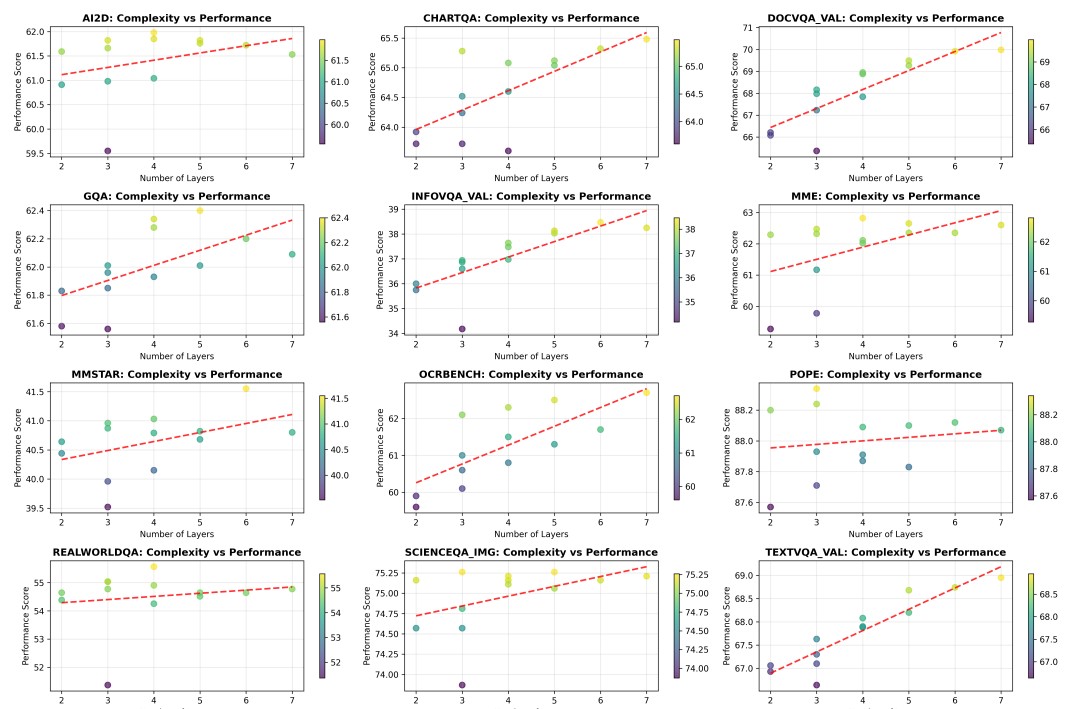

Figure 8: Performance for various VoD configurations.

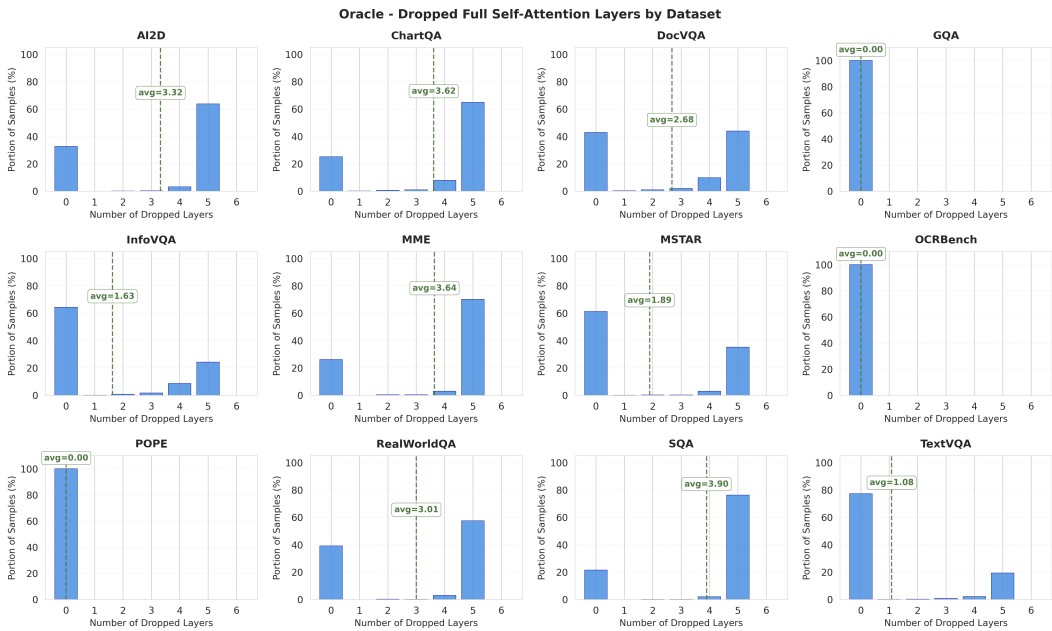

Figure 9: VoD oracle: smallest amount of layers that maximizes accuracy.

using configurations with very few self-attention layers, while hard tasks require more layers for optimal performance.

To find the optimal configuration per sample, we compare the per-config generations token by token to the tokenized ground-truth, and score it as the number of matches up to the first incorrect match.

Then, we select the configuration with the most matches. If multiple ones have the same score, we select the one with the minimum number of SA layers (i.e., the one with the most dropped layers).

## A.5 REDUCED NUMBER OF TOKENS VS REDUCED ATTENTION

In the main paper, we demonstrated that intermittent attention (VoD) is effective for hard tasks and orthogonal to token reduction methods. Here, we conduct a head-to-head comparison of these two paradigms under a similar FLOPs reduction rate of approximately $16\times$. We use our VoD-TR variant and compare it against two token reduction methods, $M^3$ and VisPruner. To ensure a fair comparison, all models were finetuned end-to-end using the same training procedure with the only difference being how the amount of input vision tokens is reduced. As shown in Table 6, while all methods perform well on easy datasets, our approach maintains a significant performance advantage on harder tasks, highlighting the benefits of preserving a larger visual context over aggressive token reduction.

Table 6: Comparison between token reduction vs intermittent attention (VoD).

| Method | Easy | | | | | | | | | Hard | | | | | Avg. FLOPs Savings |
| | RealWorldQA | SQA | GQA | MME | MSTAR | POPE | TextVQA | AI2D | Avg. (Easy) | ChartQA | OCRBench | InfoVQA | DocVQA | Avg. (Hard) | |
|---|---|---|---|---|---|---|---|---|---|---|---|---|---|---|---|
| LLaVA-OV (Li et al., 2024) | 54.0 | 67.2 | 58.3 | 60.6 | 40.6 | 88.4 | 66.0 | 56.7 | 61.5 | 60.9 | 58.8 | 40.0 | 68.7 | 57.1 | 1.0× |
| + $M^3$ | 53.6 | 75.1 | 59.1 | 63.6 | 41.5 | 88.1 | 65.5 | 62.2 | 63.6 | 62.9 | 54.0 | 34.7 | 58.0 | 52.4 | 16.0× |
| + VisPruner train. | 53.1 | 69.2 | 56.7 | 60.9 | 36.9 | 86.7 | 55.4 | 56.3 | 59.4 | 52.4 | 44.2 | 28.2 | 52.9 | 44.4 | 16.0× |
| VoD-TR (**Ours**) | 55.4 | 75.4 | 60.7 | 59.5 | 38.5 | 87.7 | 67.4 | 61.9 | 63.3 | 65.3 | 60.7 | 37.4 | 67.7 | 57.8 | 18× |

## A.6 ROUTING MECHANISM GENERALIZATION

As described in Sec. 4.3, the internal routing mechanism responsible for deciding the optimal configuration of self-attention layers to be used for each sample is trained using offline, per-dataset labels extracted from our training set. In this subsection, in order to investigate how the router performs on unseen data, we train it while excluding from its train set three datasets (AI2D, DocVQA, and GQA), and evaluate it on vision-language benchmarks including those datasets. In Table 7, we present the outcomes of this experiment (VoD-TR-excl) and contrast them with the router when trained on the full train set (VoD-TR). Our results demonstrate that, even though, naturally, the model behavior is changed, this does not lead to a drop of performance in general or in the excluded datasets in particular, thereby indicating that VoD is robust and can handle samples outside the train set's distribution.

## A.7 ADDITIONAL COMPARISON WITH THE STATE-OF-THE-ART

In the main manuscript, we compared our method against state-of-the-art approaches using a shared LLaVA-OV (0.5B) backbone. Herein, we provide additional results using a larger LLaVA-OV (1.5B) backbone. As no official 1.5B variant is openly available, we re-trained it ourselves fully using the same procedure as described in Li et al. (2024). Additionally, we've also re-implemented the baselines under the same unified setting. As the results from Table 8 show, the same conclusions hold and our method continues to outperform existing approaches on the *hard* partition of the dataset, while providing significant efficiency gains.

## A.8 ADDITIONAL DETAILS REGARDING TOKEN PACKING

To further enhance and validate the efficacy of our approach in the main section, we introduce a light adaptation for token packing, capable of working at non-power-of-two reduction rates.

Specifically, after the vision encoder, we reshape the patch embeddings back to their 2D spatial grid, interpolate the grid by a factor of $1/\sqrt{2}$ along each spatial dimension for a $2\times$ compression ratio, and then apply a space-to-depth transformation (pixel shuffle). This deterministically halves the

Table 7: Results for the routing mechanism trained on the full train set (VoD-TR), contrasted with training excluding samples from AI2D, DocVQA and GQA (VoD-TR-excl).

| Method | RWQA | SQA | GQA | MME | MMSTAR | POPE | TxtVQA | AI2D | CQA | OCRB | InfoVQA | DocVQA |
|---|---|---|---|---|---|---|---|---|---|---|---|---|
| VoD-TR | 55.4 | 75.4 | 60.7 | 59.5 | 38.5 | 87.7 | 67.4 | 61.9 | 65.3 | 60.7 | 37.4 | 67.7 |
| | 24× | 16× | 24× | 17.7× | 17.7× | 24× | 14.5× | 24× | 16× | 15× | 12× | 12× |
| VoD-TR-excl | 55.5 | 75.6 | 61.6 | 59.2 | 38.3 | 88.3 | 66.9 | 62.0 | 65.0 | 61.1 | 37.2 | 67.6 |
| | 14.5× | 15.5× | 9.8× | 11.4× | 20× | 9.8× | 12× | 14.5× | 16× | 14.1× | 12× | 12× |

Table 8: Comparison with state-of-the-art models on various vision-language benchmarks using a LLaVA-OV 1.5B backbone. The metrics used are accuracy for most datasets, except for MME where we report a score (higher is better). MME values are divided by 20.

| Method | Easy | | | | | | | | | Hard | | | | | Avg. FLOPs Savings |
| | RealWorldQA | SQA | GQA | MME | MMSTAR | POPE | TextVQA | AI2D | Avg. (Easy) | ChartQA | OCRBench | InfoVQA | DocVQA | Avg. (Hard) | |
|---|---|---|---|---|---|---|---|---|---|---|---|---|---|---|---|
| LLaVA-OV (Li et al., 2024) | 57.3 | 75.6 | 58.6 | 63.7 | 40.5 | 88.1 | 69.9 | 60.3 | 64.3 | 64.2 | 60.3 | 46.7 | 76.6 | 62.0 | 1.0× |
| VisionZip (Yang et al., 2025b) | 52.7 | 74.8 | 54.2 | 64.5 | 38.3 | 86.2 | 54.9 | 57.2 | 60.4 | 37.8 | 33.8 | 26.3 | 44.7 | 35.7 | 5.7× |
| VisionZip† (Yang et al., 2025b) | 54.6 | 76.2 | 57.2 | 63.9 | 38.1 | 87.0 | 62.0 | 58.2 | 62.2 | 50.0 | 44.3 | 30.8 | 55.1 | 45.1 | 5.7× |
| PyramidDrop (Xing et al., 2025) | 56.7 | 76.0 | 55.9 | 63.7 | 37.5 | 87.2 | 60.3 | 62.1 | 59.1 | 47.9 | 31.9 | 34.2 | 54.5 | 42.1 | 4.6× |
| VisPruner (Zhang et al., 2025b) | 52.9 | 76.0 | 53.4 | 62.3 | 38.4 | 83.3 | 54.8 | 57.5 | 59.8 | 40.0 | 34.3 | 28.4 | 51.2 | 38.5 | 5.7× |
| HiRED (Arif et al., 2025) | 54.8 | 76.0 | 56.3 | 63.2 | 39.3 | 85.9 | 59.8 | 58.2 | 61.7 | 44.1 | 42.0 | 27.6 | 52.3 | 41.5 | 5.0× |
| VoD (Ours) | 57.5 | 84.7 | 62.5 | 68.7 | 44.9 | 88.6 | 68.6 | 71.7 | 68.3 | 68.3 | 63.7 | 44.4 | 75.3 | 62.9 | 6.6× |

number of visual tokens with minimal information loss and no added parameters, complementing the computational savings from our sparse attention design. Note that the interpolation factor can be adjusted upwards to accommodate arbitrary reduction rates.

## A.9 RE-IMPLEMENTATION OF BASELINES

In this section, we detail our re-implementation of the baselines we compare with, along with the hyperparameters used. Since LLaVA-OV uses a SigLIP-400M vision encoder that does not have a CLS token, for all methods that rely on the CLS token's attention scores to select important tokens, we instead use the average attention each token receives from all other tokens in the sequence, as proposed by Yang et al. (2025b).

**VisionZip.** VisionZip (Yang et al., 2025b) is a token reduction method that selects the most important tokens, named *dominant tokens*, using the visual encoder's attention scores. To avoid losing information, the remaining tokens are merged into *contextual tokens* based on semantic similarity. In our experiments, we adapt the official VisionZip LLaVA-Next (Liu et al., 2024b) code to LLaVA-OV. Unlike LLaVA-Next, the LLaVA-OV image processing AnyRes strategy applies bilinear interpolation when the number of tokens exceeds a threshold. To prevent errors from interpolating removed tokens and to stay close to the original design, we remove this step. Yang et al. (2025b) also introduce VisionZip†, a trained version that fine-tunes the cross-modality projector. For a fair comparison, we train VisionZip† on the LLaVA-OV Single-Image 3.2M dataset (Li et al., 2024). In all VisionZip and VisionZip† experiments, we set the number of retained tokens per patch to 128, split into 104 *dominant tokens* and 24 *contextual tokens*.

**VisPruner.** VisPruner (Zhang et al., 2025b) is a training-free token pruning method that retains visual tokens based on visual attention scores. It first selects the *important tokens*, i.e., those with the highest scores, and then removes duplicates by keeping only *diverse tokens* based on their similarity. As with VisionZip, we adapt the official VisPruner LLaVA-Next code to the LLaVA-OV backbone. In our experiments, we retain 128 tokens per patch, split into 96 *important tokens* and 32 *diverse tokens*.

**HiRed.** High-Resolution Early Dropping (HiRed) Arif et al. (2025) is a plug-and-play token-reduction method designed to work under a fixed budget. It targets high-resolution LVLMs (i.e., LLaVA-Next) and drops tokens before they reach the LLM. The key idea is to evaluate the visual content of each image patch using the attention scores of the full image, then assign a budget to each patch accordingly. Within each patch, the most informative tokens are kept and passed to the LLM. In our experiments, we adapt the official code to LLaVA-OV, following the same approach used for

VisionZip and VisPruner. In our experiments, we use the same hyperparameters as the original implementation, setting a token budget of 20%.

$\mathbf{M^3}$. Matryoshka Multimodal Models ($\mathbf{M^3}$) (Cai et al., 2025) represent visual content as a nested set of tokens capturing information at different levels of detail, from coarse to fine. The visual tokens from the encoder are grouped into several coarse-to-fine levels, where the coarser tokens $X_{S_{i-1}}$ are obtained from the finer tokens $X_{S_i}$ using average pooling. $\mathbf{M^3}$ does not add any extra parameters. For a fair comparison, we train $\mathbf{M^3}$ on the LLaVA-OV Single-Image 3.2M dataset (Li et al., 2024), updating both the vision encoder and the LLM weights. In our experiments, we define a set of scales $\{X_{S_i}\}_{i=1}^M$ that reduce the number of visual tokens by factors of 1, 4, 8, and 16. For a fair comparison with our method, we report the results at an $8\times$ reduction.

**PyramidDrop**. PyramidDrop (Xing et al., 2025) is a progressive token pruning method that gradually reduces the number of visual tokens as the LLM depth increases. Specifically, the LLM layers are split into stages, and at the beginning of each stage, the number of visual tokens is reduced based on a predefined set of reduction rates, which are defined on a per-stage basis. At each pruning step, the input features are split into visual and text features, and then from the text features, only the features corresponding to the position of the last token of the user's query or instruction are kept, resulting in $N_v \times d$ visual features and a single text feature vector. Then, the next attention layer to be executed is applied over these features, with the text features as the query, resulting in image-text attention weights that are interpreted as a per-visual token importance score. These scores, together with a per-stage pre-defined drop-rate, are used to keep the top-k scoring visual tokens, with k as the target visual token to keep. This procedure is applied progressively throughout the LLM's depth at the beginning of each stage. In our implementation, for LLaVA-OV 0.5B with a 24-layer LLM, we used 4 stages defined as $(1, 2-6, 7-12, 13-24)$ with drop-rates of $(1.0, 0.3, 0.2, 0.1)$ for easy datasets, and $(1-4, 5-10, 11-16, 17-24)$ with drop-rates of $(1.0, 0.5, 0.25, 0.125)$ for hard datasets with average FLOPs saving of $4.2\times$. As for LLaVA-OV 1.5B with a 28-layer LLM we used 5 stages defined as $(1, 2, 3-6, 5-10, 11-28)$ with drop-rates of $(1.0, 0.5, 0.3, 0.2, 0.1)$ for easy datasets, and $(1-2, 3-8, 8-12, 13-18, 19-28)$ with drop-rates of $(1.0, 0.75, 0.5, 0.25, 0.125)$ for hard datasets with an average FLOPs saving of $4.6\times$.

## A.10 ADDITIONAL DETAILS ON CENTERED KERNEL ALIGNMENT (CKA)

In Figure 3, we show how vision features evolve across layers within the LLM transformer of the LVLM by computing the Centered Kernel Alignment (CKA) (Cortes et al., 2012). More specifically, to compute it, let $X \in \mathbb{R}^{n \times d}$ and $Y \in \mathbb{R}^{n \times d}$ represent the vision features extracted from two different layers, where $n$ is the number of tokens and $d$ is the feature dimension. The CKA computation begins by forming the Gram matrices $K = XX^T$ and $L = YY^T$. These matrices are then centered using the centering matrix $H = I_n - \frac{1}{n}1_n$, where $I_n$ is the identity matrix and $1_n$ is an $n \times n$ matrix of ones. The centered Gram matrices are given by $\tilde{K} = HKH$ and $\tilde{L} = HLH$. The CKA similarity between $\tilde{K}$ and $\tilde{L}$ can then be computed as:

$$\text{CKA}(\tilde{K}, \tilde{L}) = \frac{\langle \tilde{K}, \tilde{L} \rangle_F}{\|\tilde{K}\|_F \|\tilde{L}\|_F}, \tag{4}$$

where $\langle \cdot, \cdot \rangle_F$ denotes the Frobenius inner product, and $\| \cdot \|_F$ represents the Frobenius norm.

