# OpenReview forum: "Vision-on-Demand: Efficient Visual Language Understanding with Intermittent Attention"
_ICLR.cc/2026/Conference — ICLR 2026 Conference Withdrawn Submission_

### Official Review · Reviewer_rsmL · 2025-10-27

**Soundness:** 3
**Presentation:** 2
**Contribution:** 3
**Rating:** 4
**Confidence:** 5

**Summary:**

The paper targets the efficiency of vision-language models, particularly addressing the high number of visual tokens. Its main approach is to minimize processing on visual tokens within the LLM decoder. First, it argues that for “easy” VLM tasks, text tokens do not require frequent attention to visual tokens. Second, using CKA analysis, it shows that visual tokens for “easy” VLM tasks do not change significantly across the LLM layers, leading to the conclusion that self-attention among visual tokens is only necessary for more challenging VLM tasks. Based on these observations, the paper proposes the Vision-On-Demand (VoD) method, which applies text-to-image and image-to-image attention only in a few layers of the LLM. Experiments in the paper demonstrate more than an 8× theoretical FLOPs reduction compared to the baseline, with comparable accuracy.

**Strengths:**

- The research problem addressed by the paper (efficiency challenges caused by the high number of visual tokens) is crucial for practical applications of vision-language models.
- The observations presented in the paper are interesting:
  - Text-to-image attention is sparse, and for some easier tasks, it is minimal, suggesting that such attention is not needed throughout the network.
  - The original visual features produced by the vision encoder remain largely unchanged for some easier tasks, indicating that image-to-image attention is necessary only for certain tasks and in a few layers.
- The proposed method, Vision-On-Demand (VoD), is well-motivated and intuitive.
- The results show a significant FLOPs reduction while maintaining accuracy.
- For a consistent comparison, different baselines are reimplemented to eliminate other confounding effects.

**Weaknesses:**

- One major issue with the paper is its positioning relative to baselines. Throughout the paper, the number of visual tokens ($N_v$) and text tokens ($N_t$) is not mentioned. What input resolution is considered for the LLaVA-OV baseline [1]? Are you using a full $3\times3+1=10$ patch configuration, equivalent to $10\times729=7290$ visual tokens? This seems like an extreme baseline to report all FLOPs savings against. More recent works, such as [2] (not discussed in the paper), use the same Qwen LLM as this work but produce significantly fewer visual tokens (e.g., 256 instead of 7290). It remains unclear whether the proposed method benefits efficient vision-language models such as [2].
- The presentation in Sections 4.2.3 and 4.3 is confusing and incomplete. Many important details are moved to the appendix, while the main paper focuses on less critical information. For example, how pseudo-labeling is done is part of the proposed algorithm but is missing from the main text, whereas dataset mixtures and training hardware details are included. A major reorganization is needed to improve the presentation of the main algorithm. See detailed comments below:
  - It would be clearer if a “configuration” were defined first. Perhaps as the arrangement of CA and SA layers in a given LLM architecture.
  - Section 4.2.3, Step 1: What does “maximum number of cross-attention” mean? Does it refer to an optimal number for achieving good accuracy?
  - Section 4.2.3, Step 1: It is mentioned that $L_{CA} = L_{SA} = L/3$ was found empirically. However, experiments supporting this observation are neither included nor referenced.
  - Section 4.2.3, Step 2: This step is not clearly defined, and supporting experiments are not included or referenced.
  - Section 4.2.3, Step 3: What does “randomly selecting” mean? Does it imply that for each optimization step a new random configuration is sampled? Please clarify.
  - Section 4.2.3 step 3: What do “these viable configurations” refer to? Are these configurations identified in Stage 2? Please clarify.
  - Section 4.3: It is unclear what exactly is logged. For a dataset with $N$ samples and a model with $L$ layers, what must be logged? What is the final pseudo-label? Does it represent a layer configuration per sample? How many inference runs per sample are required to determine this? Please clarify.
  - Section 4.3: The paper mentions that directly learning the router is unstable. Are there experiments demonstrating this? Please reference and explain.
- Results in Table 3 (upper part) show insensitivity to the number of cross-attention layers, but it is also argued that accuracy saturates at 8 CA layers. Please clarify and include results for fewer CA layers, including 0.
- The paper briefly mentions actual runtime latency at the end of Section 6, yet this is a critical metric. Please include an accuracy vs. time-to-first-token plot for baselines and different VoD variants, including VoD-TR, across various compression ratios.

Minor issues:
- Lines 265–269 discuss the choice of positional encoding via convolution without sufficient detail. This paragraph seems out of place and unrelated to the main contribution. It could be moved to the appendix.
- Lines 478–479 contain typos: “VoD uses” and “VoD trains.”

References:
[1] Li, Bo, et al. “LLaVA-OneVision: Easy Visual Task Transfer.” arXiv preprint arXiv:2408.03326 (2024).
[2] Vasu, Pavan Kumar Anasosalu, et al. “FastVLM: Efficient Vision Encoding for Vision-Language Models.” Proceedings of the Computer Vision and Pattern Recognition Conference. 2025.

**Questions:**

- Figure 1: For the proposed method (VoD), the actual FLOPs depend on each sample due to routing. Are you showing averaged FLOPs reduction rates? If so, why are they the same for both the Hard and Easy datasets?
- Are all results in the paper, except those in Table 4, based on the "Universal model" when referring to VoD and VoD-TR?
- What model is used for the analysis in Figure 2?
- Figure 2: How is the attention score calculated? Is it averaged over all heads per layer? Also, is it averaged over all samples in the corresponding dataset? If so, what is the variance? Does the mentioned “saw-tooth” pattern appear for all heads in the model?
- What does the distribution in Figure 4a represent? Please provide details of the random visual token dropout process. Additionally, annotating tasks as Hard/Easy in Figure 4a would clarify the basis of this categorization.
- What is the significance of the result shown in Figure 4b? It is not discussed in the text.
- In line 359, is the vision encoder (SigLIP) frozen during stages 1 and 2?

---

### Official Review · Reviewer_J33q · 2025-10-29

**Soundness:** 2
**Presentation:** 2
**Contribution:** 3
**Rating:** 4
**Confidence:** 2

**Summary:**

This work proposes Vision-on-Demand (VoD), a method that accelerates inferences for VLMs by sparsifying the image-text and image-image interactions without token dropping. It incorporates cross-attention layers to allow text tokens to gather information from the image tokens, further full self-attention layers are enabled to allow information refinement. With evaluations on the LLaVA-OV model, the proposed method is shown to obtain better reults compared to other acceleration methods.

**Strengths:**

- The idea of removing the inter-connection of image and text tokens is interesting for accelerating VLMs.
- The proposed method is shown to obtain good results on the target tasks.

**Weaknesses:**

- It seems that only LLaVA-OV is utilized for evaluation, while more other models (such as Qwen VL series) should also be included to show that the proposed method can work well for different models.
- The proposed method changes the model architecture and needs to be trained. It remains a question how well it can generalize to different scenarios, since the training usually requires a specific training set.
- I'm wondering how much the savings on FLOPs can be transformed to the final latency improvement. There is a lack of evaluation on this aspect.

**Questions:**

(Please refer to the weakness part.)

---

### Official Review · Reviewer_jbSU · 2025-10-31

**Soundness:** 2
**Presentation:** 3
**Contribution:** 2
**Rating:** 4
**Confidence:** 4

**Summary:**

This paper introduced a new method Vision-on-Demand (VoD) for building efficient vision language models. Starting from the observation that for different tasks, the amount of compute and interaction required for visual tokens are very different, the authors proposed a new VLM architecture consists of regular LLM layers, cross attention layers and self-attention layers (the vision encoder and project remains the same). The cross attention layers allows text tokens to attend to visual tokens, and self-attention layers allow bidirectional interaction between text and visual tokens for better feature update. Additionally, the model is trained with a internal routing mechanism where a special token is used to predict the optimal inference configuration. The model is trained in a similar way as LLaVA-OV and architecture configurations are dynamically sampled during training to ensure the model can handle different inference conditions.

**Strengths:**

The method is grounded on a very practical but often overlooked aspect: that there is no universal visual token compression methods, hence an adaptive inference strategy should delivery optimal performance. The proposed VoD method addresses this issue by allowing the model to select optimal inference path for every example.

Compared to LLaVA-OV baseline and other visual token pruning methods, VoD achieves salient performance gains on both easy and hard tasks and have much better FLOPs savings, showing its effectiveness. Moreover, the VoD are orthogonal to visual token pruning methods, and it can be combined with those methods to further save computes.

**Weaknesses:**

My main concern is about the generality of VoD. The author mentions that VoD has self-attention layers, which allow image-to-image, text-to-image, image-to-text and text-to-text interactions, meaning that this is bidirectional attention without causal mask. This means that for every new token being generated, all token’s self-attention will need to be recomputed. When the generation becomes long, e.g. long CoT settings, this would incurs a very large computation overhead. Thus I’m not sure if this method can actually save computation in general.

The authors mentioned that both cross-attention and self-attention layers are inserted into the LLM. Per my understanding, this adds additional layers/parameters to the model. In other words, VoD actually has more parameters and layer depth than LLaVA-OV, despite the fact that vision tokens are not passing through LLM layers. This would definitely give VoD an unfair advantage (large modeling capacity) compared to baseline methods, thus the results and gains on those benchmarks should also be taken with a grain of salt.

**Questions:**

Can authors clearly explain how much more parameters/layers does VoD has, and how much more compute would it require when generating sequences of different lengths? (e.g. 1k, 2k, 4k, 8k, etc)

---

### Official Review · Reviewer_MnZf · 2025-11-02

**Soundness:** 3
**Presentation:** 3
**Contribution:** 3
**Rating:** 4
**Confidence:** 3

**Summary:**

This paper proposes Vision-on-Demand (VoD), a new paradigm that does not reduce visual tokens but allocates computations on demand in large visual language models (LVLM) through the "intermittent attention" mechanism: Provide static visual context for the text with a small amount of lightweight cross-attention layers, only insert self-attention layers at key layers to refine high-resolution visual tokens, and train a unified network to support different computing power budgets. Then, a lightweight policy network is introduced to dynamically select the number of self-attention layers based on sample complexity, achieving sample-level adaptive reasoning. VoD is orthogonal to the existing token compression methods and can be superimposed to further accelerate. Experiments show that while VoD achieves SOTA accuracy on multiple visual language benchmarks, its reasoning FLOPs are reduced by 8.6 to 18×, especially showing significant advantages in difficult tasks with high resolution and fine-grained visual understanding.

**Strengths:**

- This paper points out that the existing "visual token compression" methods experience a sharp decline in performance due to information bottlenecks in fine-grained tasks. For the first time, it considers the efficiency issue from a brand-new perspective of "retaining all high-resolution visual information and only sparsity the computing layer", providing an orthogonal direction distinct from the mainstream for the efficient utilization of LVLM.

- This method decouples the computing layer into "low-cost cross-attention (read-only visual context) + a small amount of self-attention (on-demand refinement of visual tokens)", and train a unified network in conjunction with a lightweight routing network to achieve samply-level dynamic layer selection. This not only significantly reduces FLOPs without losing information but also seamlessly superimposes with existing token compression, balancing accuracy and efficiency

**Weaknesses:**

- The method requires repeatedly inserting and skipping self-attention and cross-attention layers within the LLM, and training additional routing networks to determine the number of layers per sample, which leads to complex implementation, high engineering deployment difficulty, and the need for fine-tuning of the training and inference processes, significantly increasing the system implementation and maintenance costs.

- Could the proposed method adapt on inference frameworks like vllm, sglang? How to establish it on flash attention?

- The models used for experiments is too small. It would be better to use larger models over 7B parameters.

**Questions:**

See above.

---

### Note · Authors · 2025-11-13

I have read and agree with the venue's withdrawal policy on behalf of myself and my co-authors.